# SpatialSSL: Whole-Brain Spatial Transcriptomics in the Mouse Brain with Self-Supervised Learning

**Till Richter**
Helmholtz Munich
Technical University of Munich
till.richter@helmholtz-munich.de

**Anna Schaar**
Helmholtz Munich
Technical University of Munich
anna.schaar@helmholtz-munich.de

**Francesca Drummer**
Helmholtz Munich
Technical University of Munich
francesca.drummer@helmholtz-munich.de

**Cheng-Wei Liao**
Technical University of Munich
c.liao@tum.de

**Leopold Endres**
Technical University of Munich
leopold.endres@tum.de

**Fabian Theis**
Helmholtz Munich
Technical University of Munich
fabian.theis@helmholtz-munich.de

## Abstract

Self-supervised learning (SSL) is a rich framework for obtaining meaningful data representations across large datasets. While SSL shows impressive results in computer vision and natural language processing, the single-cell field's diverse applications still need to be explored. We study SSL for the application of cell classification in cellular neighborhoods of spatially-resolved single-cell RNA-sequencing data. To address this, we developed an SSL framework on spatial molecular profiling data, integrating a cell's molecular expression and spatial location within a tissue slice. We demonstrate our methods on a large-scale whole mouse brain atlas, recording the gene expression measurements of 550 genes in 4,334,174 individual cells across 59 discrete tissue slices from the entire mouse brain. Our empirical study suggests that SSL improves downstream performance, especially in the presence of class imbalances. Notably, we observe a more substantial performance improvement on the sub-graph level than the full-graph level.

## 1 Introduction

Single-cell genomics and spatial transcriptomics have enriched our understanding of biological systems, providing detailed insights into cellular diversity and function Fischer et al. [2023], Palla et al. [2022], Larsson et al. [2021], Zhuang [2021]. While techniques like single-cell RNA sequencing (scRNA-seq) and single cell Assay for Transposase-Accessible Chromatin using sequencing (scATAC-seq) excel in molecular profiling, spatial methods like MERFISH Chen et al. [2015] and Stereo-seq Chen et al. [2022] add an insightful spatial dimension. These technologies offer a multi-dimensional view of biological systems, capturing molecular states and spatial coordinates. This enables insights into tissue states and local cellular micro-environments, that relate to biological functions Fischer et al. [2023]. However, as datasets grow in size and complexity, there is a pressing need for new machine learning techniques that can effectively use unlabeled data for various downstream applications.

NeurIPS 2023 AI for Science Workshop.

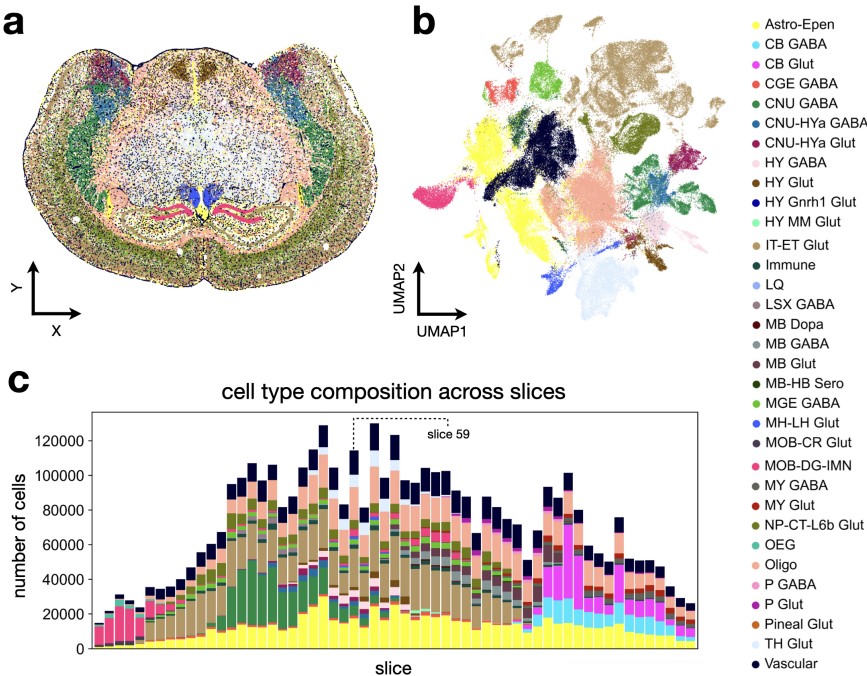

Figure 1: The multiple facets of the whole adult mouse brain brain atlas Zhang et al. [2023], focusing on the spatial distribution, clustering of the molecular features, and cell type composition across slices. (a) Proximity graphs in spatial transcriptomics data. Shown is the spatial allocation of slice 59 with all cell types superimposed. (b) Depicts a UMAP representation of the molecular embedding of all cells in slide 59 (n = 114,396 cells) with cell-type superimposed. (c) Illustrates the distribution of cell types across the entire 59 full coronal sections spanning the entire mouse brain. The whole adult mouse brain atlas reveals unique cell type organization patterns across the different brain sections. The slices range from 14,878 to 130,112 cells, with an average cell count of 62,342 per slice.

Self-supervised learning (SSL) has shown promise in fields like computer vision Bardes et al. [2022], Chen et al. [2020] and natural language processing Radford et al. [2018], Devlin et al. [2018], often setting new performance benchmarks. However, its potential in single-cell genomics is still largely unexplored. Unlike traditional methods focusing on specific tasks or limited datasets, SSL aims to build a more general model applicable across different tasks Weng and Kim [2021], Balestriero and LeCun [2022]. This paper addresses this gap by developing SSL methods designed explicitly for spatially resolved single-cell data, offering a solution to its unique computational challenges.

This paper introduces SpatialSSL[1], a novel framework tailored for SSL on spatially resolved single-cell datasets. Our primary contribution lies in developing and validating SSL algorithms optimized for the unique challenges presented by high-dimensional, sparse and spatially distributed data. We empirically demonstrate the efficacy of SpatialSSL through testing on the cell type annotation task, utilizing the novel whole-brain atlas dataset BICCN 2.0 Zhang et al. [2023][2] as our evaluation platform. Exploring SSL in the context of spatial data serves as a step toward large, foundational models capable of integrating graph-based and non-graph-based data of the single-cell field.

# 2   Data

## 2.1   Spatial whole adult mouse brain atlas (BICCN 2.0)

Our empirical evaluation of self-supervision in spatial trancriptomics is based on a high-resolution spatially resolved atlas of whole adult mouse brain Zhang et al. [2023]. The atlas is part of the

---

[1]Code is available at: github.com/theislab/spatial_atlas_ssl
[2]Data publicly available at Zeng et al. [2023]

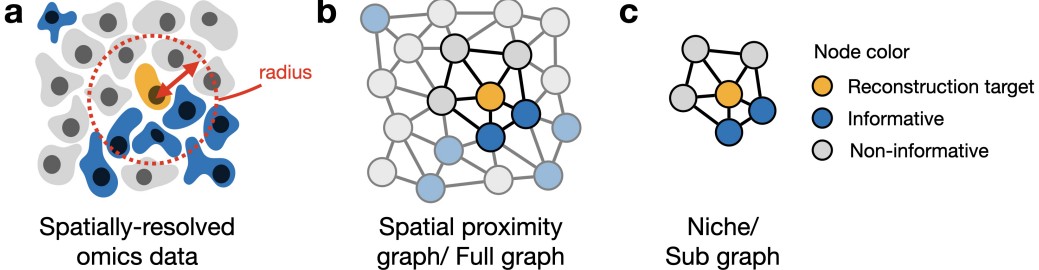

Figure 2: Comparison of data representation approaches. Figure (a) illustrates exemplary spatial data in which a cell's (a node's) neighborhood is defined by a radius of euclidean distance. Figure (b) portrays the data representation as a full graph, containing all cells of the sample. Figure (c) shows the representation as sub graphs, containing a subset of nodes that are relevant to the node of interest. We hypothesize that SSL struggles with non-informative neighbors and large feature spaces in full-graphs, while sub-graphs offer a more favorable setting for meaningful data representation.

BRAIN Initiative Cell Census Network (BICCN) aiming to generate cell type atlases for mouse and human brain. The spatially resolved transcriptomic dataset was generated with MERFISH Chen et al. [2015], a spatially resolved transcriptomic method, and captures the gene expression profile of 550 genes across 4,334,174 individual cells from one entire whole adult mouse brain, distributed over 59 serial full coronal sections.

All cells available in the atlas are segmented and mapped to a common coordinate framework to obtain precise spatial coordinates for each individual cell across the different brain sections. The spatial locations of each cell are provided as X and Y coordinates within the atlas.

Additionally, the atlas provides a unique hierarchical cell type annotation with different levels of granularity, spanning from seven divisions up to 5,200 identified clusters. One can visualize the spatial organization of the different annotation levels per section within the atlas to observe the spatial organization of different sub-states in the brain (Figure 1a).

Datasets obtained with single-cell sequencing technologies are typically high-dimensional objects with many measured cells and genes. To assess the similarity of cells with respect to their molecular features, one can embed the data into a lower-dimensional representation to identify the underlying data topology. We applied the Uniform manifold approximation and projection (UMAP) algorithm on one section of the whole adult mouse brain atlas to visualize the different cell types present in the brain region (Figure 1b). As we can observe, cell sub-types cluster together and express distinct molecular states.

We additionally inspect the cell type composition across the entire atlas and the 59 distinct sections. The different brain sections spanning from dorsal to ventral parts of the brain show a diverse cell type organization (Figure 1c).

## 2.2 Spatial graphs of cells

Spatial transcriptomics data resolves cellular measurements at distinct spatial locations. This data can be encoded as spatial graphs of cells which are computed from the spatial molecular profiling data and the cell-wise spatial coordinates (Figure 2a). The graph is assembled based on spatial proximity with respect to euclidean distance in the two-dimensional space. We calculate an adjacency matrix indicating the spatial connectivities of individual cells. One can compute the spatial proximity graph on the entire capture area or on each section separately. We furthermore refer to the entire tissue graph as full graph (Figure 2b). Additionally, biological insights can stem from the connectivities of individual target cells. The target cell and all it's neighbours form a niche or cellular microenvironment that can be anaylzed in various ways. We further refer to the niche as sub-graph (Figure 2c).

(For experimental details, see A.1).

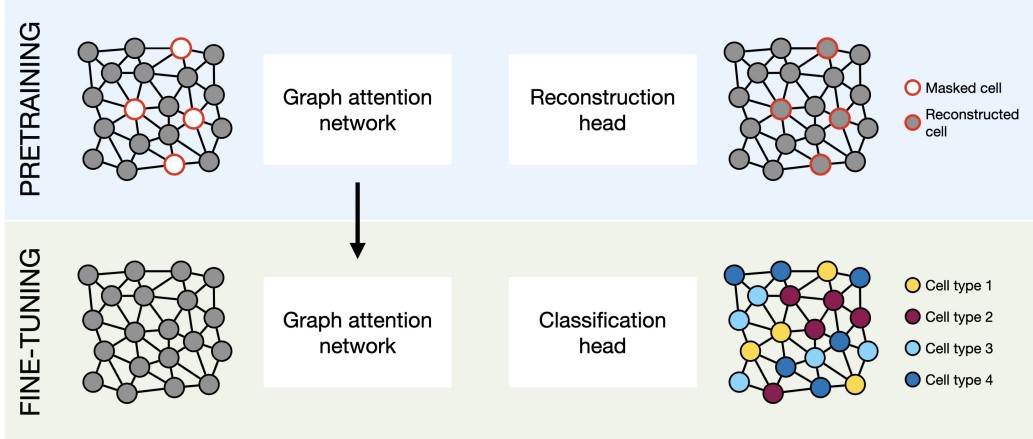

Figure 3: Visualization of our self-supervised learning experiment. A graph, in which nodes are cells and edges are constructed via their euclidean distance on the image, serves as input data. During pretraining, the graph attention network reconstructs masked inputs. After transferring the model weights, the fine-tuning step comprises of another graph attention network and subsequent classification used for cell type annotation.

We adopt two divergent strategies for sample creation:

**Full-Graph**: This variant comprises each image (section) to one graph, resulting in 59 individual graphs. The individual graphs per section are not connected to other graphs in the dataset. Full graphs can capture globale dependencies across the sections and can serve as a comprehensive basis for pretraining graph models.

**Sub-Graph**: This variant creates multiple smaller graphs - sub-graphs - per image (section). Thereby, the focus lies localized interactions by creating EgoNet sub-graphs for each cell, where EgoNet is a type of sub-graph centered on a particular node and its neighborhood. Defined by a center node and its k-hop neighbors, these sub-graphs offer a more granular view, potentially reducing noise and improving model performance.

By structuring the data as graphs, we enable graph-based neural networks to exploit the spatial relationships between nodes for more accurate feature or class prediction. The distinction into full- and sub-graph aims to evaluate the hypothesis of diminishing effects of self-supervised learning with increasing feature spaces. The rationale behind this idea is that larger feature spaces include more uninformative features for pairwise relationships, and that the applied data augmentation has less relative weight.

## 3 Self-Supervised Learning on Spatial Data

### 3.1 Self-Supervised Learning

Self-supervised learning follows the idea that data and pairwise relationships are sufficient to learn meaningful data representations without the need for explicit labels Balestriero and LeCun [2022]. Thereby, the model learns to distill meaningful signals from noise Von Kügelgen et al. [2021]. This paradigm is compelling to mitigate challenges of large, real-world datasets, such as class imbalances Liu et al. [2021]. The SSL framework consists of a pretraining phase, where the model learns to extract features from the data, and a finetuning phase, where these learned features are applied to a specific downstream task (see Figure 3). The applicability of the data representation to improve multiple downstream tasks makes self-supervision particularly interesting for real-world data. In this study, we use mask inpainting as pretraining task, inspired by the efficacy of masked autoencoders He et al. [2022], Cao et al. [2022]. Specifically, we employ a graph attention network to mask and reconstruct masked gene expressions. Given a binary mask $M$, the model $f_\theta$ is trained to reconstruct the masked components of the input $X$ using the unmasked components as context, i.e.:

$$\hat{X}_{\text{pred}} = (\mathbf{1}_{m \times n} - M) \odot \hat{X} \tag{1}$$
$$= (\mathbf{1}_{m \times n} - M) \odot f_\theta(M \odot X) \tag{2}$$

where $\mathbf{1}_{m \times n}$ is a matrix of ones with the same dimensions as $M$. The mask $M$ allows task-specific design. This work introduces a mask $M$ generated through full-feature masking, targeting random nodes within the graph. Specifically, all features of a selected node are masked, impacting a random subset of nodes in each graph with a patch size of 1. Translated to our application, all gene expression values of a cell are masked on a randomly selected number of cells in the graph.

### 3.2 Cell-Type Annotation in Spatially Resolved Single-Cell Genomics

Cell-type annotation in spatially resolved single-cell genomics involves classifying each node — attributed with a vector of $n$ real-valued RNA-sequencing counts — into distinct classes representing unique cellular identities. The task is challenging due to the noise and heterogeneity inherent in large-scale datasets.

**Traditional Approaches and Their Limitations**: For small datasets with well defined cell-types, simple models can perform well. For instance, linear models like Celltypist Domínguez Conde et al. [2022], based on logistic regression, have shown promising results. However, their modeling capacity is limited, making them unfavorable for large and heterogeneous datasets. Furthermore, the objective is to develop models capable of robust generalization in order to faithfully represent the data. This aim requires rigorous evaluation through performance metrics sensitive to class imbalances, such as the macro F1-score.

**The Promise of Self-Supervised Learning**: A critical aspect of robust generalization lies in the model's ability to learn universally informative features across the data manifold while avoiding the influence of technical noise and spurious correlations. Self-supervised learning methods, such as in our SpatialSSL framework, offer a promising route for achieving this. These methods aim to build rich, generalizable data representations, especially when applied to large and complex datasets like the spatial brain atlas. By doing so, the model learns to focus on invariant concepts rather than overfitting to spurious correlations in the training data, thereby promising to enhance its performance in downstream tasks such as cell type classification.

## 4 Results

Our empirical evaluation focuses on both supervised (noted as No SSL) and self-supervised learning models (noted as SSL) applied to full-graph and sub-graph representations of the data. The performance metrics employed are the micro and macro F1 scores, as illustrated in Figure 4. Notably, the macro F1 score provides a balanced measure by accounting for class imbalances, making it particularly relevant for our study.

**Self-Supervised Learning Enhances Performance**: A key observation, highlighted in the upper-left quadrant of Figure 4, is that SSL improves the performance in cell type annotation. This improvement is especially evident in the macro F1 score, a metric sensitive to class imbalances. This observation empirically supports our hypothesis that SSL enhances downstream tasks' performance and robustness.

**Sub-Graphs Outperform Full-Graphs**: The upper-right plot of Figure 4 reveals that representing the data as sub-graphs leads to better performance than full-graphs. This difference aligns with our hypothesis that sub-graphs provide a helpful environment for meaningful data representation. In a sub-graph representation, the neighbors of a node are more informative, facilitating effective learning.

**Importance of Sensitive Metrics**: The lower-left plot of Figure 4 illustrates the class imbalances inherent in the dataset. A noticeable decline in performance on sensitive metrics like the macro F1 score suggests that models are exposed to these imbalances. This performance decrease emphasizes the critical role of sensitive metrics in providing a reliable evaluation of model performance on real-world datasets.

**SSL's Differential Impact on Sub-Graphs and Full-Graphs**: Finally, the lower-right plot of Figure 4 shows that while SSL improves performance on sub-graphs, its impact is smaller on full-

| Metric | Full-Graph | | Sub-Graph | |
|--------|------------|------|-----------|------|
| | No SSL | SSL | No SSL | SSL |
| F1 Micro | 0.57 | 0.57 | 0.60 | **0.61** |
| F1 Macro | 0.54 | 0.55 | 0.55 | **0.59** |

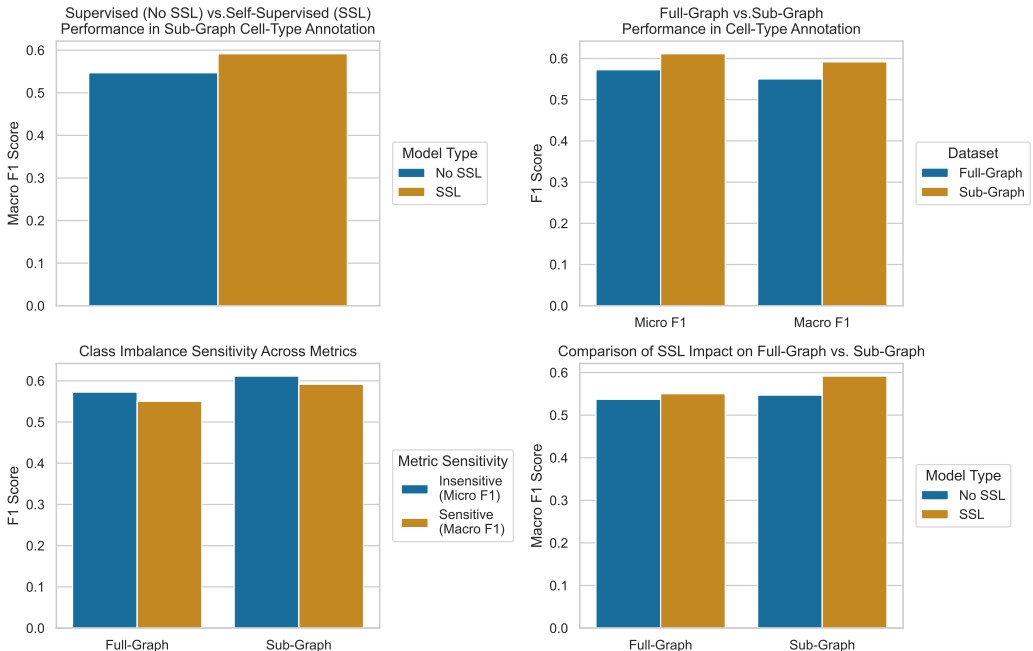

Figure 4: Cell type annotation performance on a hold-out test image. Upper panel: Comparative table of F1 micro, and F1 macro across full-graph and sub-graph data representation for supervised (No SSL) and self-supervised models (SSL). Lower panel: Corresponding barplot visualization of performance comparisons.

graphs. This observation further confirms our hypothesis that sub-graphs offer a more favorable learning environment, also for self-supervised learning. SSL has limited utility in the context of full-graphs, failing to impact performance strongly.

In summary, our results provide evidence for the efficacy of self-supervised learning in enhancing cell type annotation, particularly when applied to sub-graph representations of the data. These findings validate our initial hypotheses and underscore the importance of choosing the appropriate data representation and performance metrics. The differential impact of SSL on sub-graphs versus full-graphs raises questions about the optimal granularity for effective learning. These observations serve as a starting point for a broader discussion on the role and limitations of self-supervised learning in the context of spatially resolved single-cell genomics.

## 5 Discussion

Our empirical findings suggest that self-supervised learning improves cell type annotation performance in the context of spatially resolved scRNA-seq data. Larger performance gains are observed in the sub-graph representation, than the full-graph representation. We attribute this disparity to the inherent differences in the granularity of the feature spaces modeled by the two approaches, with the full-graph capturing much more nodes, including uninformative ones. By focusing on localized cellular interactions (niches), the sub-graph enables the self-supervised algorithms to capture biologically meaningful pairwise relationships. In contrast, the full-graph, which attempts to represent the entire

spatial data sample, may introduce complexity that hinders the effective learning of such relationships. This observation raises relevant questions about the optimal scale at which self-supervised learning should be applied in spatial transcriptomics and beyond, deserving further investigation and validation across SSL methods and datasets.

# 6    Conclusion

Self-supervised learning offers a promising avenue for extracting meaningful data representations, particularly in high-dimensional and complex datasets such as scRNA-seq and spatially resolved single-cell data. This study employed masking as the SSL paradigm to explore its efficacy on cellular graphs derived from the BICCN 2.0 dataset. Our empirical findings suggest that SSL improves the model's downstream performance in cell type annotation and robustness against class imbalances. Notably, we observe a bigger improvement at the granularity of local neighborhoods (sub-graphs) rather than the global structure (full-graphs). This divergence leads us to suspect that the efficacy of self-supervised learning in spatial transcriptomics is intrinsically tied to the scale of the neighborhood, emphasizing the need for targeted, scale-specific approaches in future research.

## Acknowledgments and Disclosure of Funding

We thank the Hongkui Zeng, Michael Kunst and the Allen Institute for Brain Science for sharing the segmented and annotated BICCN 2.0 atlas pre-publication.

TR is supported by the Helmholtz Association under the joint research school "Munich School for Data Science" (MUDS) and CausalCellDynamics (grant Interlabs-0029). ACS acknowledges support from the Bavarian Ministry of Science and the Arts in the framework of the Bavarian Research Association "ForInter" (Interaction of human brain cells) and by the Helmholtz Association through CausalCellDynamics (grant Interlabs-0029). FKD is supported by the Hertie Network of Excellence in Clinical Neuroscience. FJT reports receiving consulting fees from Roche Diagnostics GmbH and ImmunAI, and ownership interest in Cellarity, Inc. and Dermagnostix. The remaining authors declare no competing interests.

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

# A Appendix

## A.1 Experimental Details

Squidpy Palla et al. [2022] was used to construct graphs from the spatial data. Each cell is represented as a node in the graph, nodes are connected with an edge if the euclidean distance between them measured in μm is less than the preset *radius* parameter. *radius* was set to 40 μm in this study. We inferred k-hop subgraphs (hops = 2) for the Sub-Graph representation using the *k_hop_subgraph* method provided by pytorch-geometric Fey and Lenssen [2019]. After benchmarking experimental parameters, we used a masking rate of 20% (i.e., mask all features in 20% of the cells).

We used 58 slices with an 80/20 training and validation split, for both pre-training and fine-tuning processes. A single slice, not previously seen by the model, is allocated exclusively for testing to enable a robust comparative evaluation.

Hyperparamters of training and pre-training steps:

- *radius*: 40
- *k-hop*: 2
- *bottleneck*: 64
- *Model*: Graph Attention Networks
    - *hidden layers*: 4
    - *learning rate*: 0.002
    - *optimizer*: Adam

The workflow was implemented in a snakemake pipeline in SpatialSSL. It enables the user to test different configurations of the experiment. The user can train SpatialSSL using custom data. They can select the parameters of the dataset creation such as the radius which is used to infer connections between individual cells in the graph. Furthermore, the user can configure the model architecture and training setup by setting the bottleneck dimensions, the type and the number of hidden layers, the learning rate, and batch size. More information can be found in the project repository.

## A.2 Dataset Memory Usage

We adopted memory-saving methods (i.e. lower batch size, checkpoint, and sparse-tensor) to reduce memory usage on the GPU for the full-graph representation. The full-graph representation requires 22-147 GB RAM on the machine during fine-tuning with the graph attention model with 4 hidden layers. The sub-graph representation requires 7 MB for fine-tuning on each sub graph, which provides flexible model architecture design and able to accelerate the training process on the GPU with less GPU RAM.

## A.3 Test Slice Selection

The test slice was chosen based on its minimal cell-type imbalance. This was determined by calculating the average number of each cell type across all slices and selecting the slice wherein the difference in the quantity of each cell type was smallest, ensuring a representative and balanced dataset for testing purposes.

