# OpenReview forum: "SpatialSSL: Whole-Brain Spatial Transcriptomics in the Mouse Brain with Self-Supervised Learning"
_NeurIPS.cc/2023/Workshop/AI4Science — NeurIPS2023-AI4Science Poster_

### Official Review · Reviewer_6X9m · 2023-10-24
**Self-supervised learning improves GNN-based classification**

**Rating:** 6
**Confidence:** 4

**Review:**

The author suggests utilizing GNN to integrate spatial information with Merfish, representing the entire mouse brain dataset. The evaluation section highlights performance improvements in the sub-graph representation. As anticipated, the sub-graph yields superior accuracy compared to the full-graph. This is likely due to the implicit inductive bias suggesting that the local environment offers a more favorable information-to-noise ratio than the global environment in the present representation.

While this study serves as a promising starting point, it would enhance clarity if the training procedure, particularly the NO SSL part, were elaborated upon. Specifically, further details regarding the data portion used for training would be beneficial. Additionally, given the current performance constraints, the author might consider delving into improved graphical representations and refining self-supervised learning methodologies. The manuscript would be more persuading if more spatial transcriptomics methods were discussed and compared.

---

### Official Review · Reviewer_YPoe · 2023-10-25
**Verified the effectiveness of SSL in improving downstream performance for cell classification using graph representation**

**Rating:** 6
**Confidence:** 5

**Review:**

Pros
The result showing performance difference is full-graph and sub-graph is interesting.



Comments
It would be interesting to see the SSL performance on non-graph-based data representation (for cell classification)

---

### Meta-Review · Area_Chair_Bnn6 · 2023-10-27

**Recommendation:** Accept (Poster)
**Confidence:** 4

**Metareview:**

The study's exploration of performance difference between full-graph and sub-graph methods is interesting. However, the manuscript needs some revisions for improvement. It would benefit from more clarity on the training process (specifically, the NO SSL aspect) and information about the training data. Reviewers have suggested a more extensive discussion and comparison of spatial transcriptomics methods for a stronger study. They also recommend adding an analysis of SSL performance on non-graph-based data representations for cell classification. Additionally, reviewers propose exploring better graphical representations and refining self-supervised learning techniques to enhance the study's quality and impact.